# A Perspective on Therapeutic Targeting Against Ubiquitin Ligases to Stabilize Tumor Suppressor Proteins

**DOI:** 10.3390/cancers17040626

**Published:** 2025-02-13

**Authors:** Ishaar P. Ganesan, Hiroaki Kiyokawa

**Affiliations:** Department of Pharmacology, Feinberg School of Medicine, Northwestern University, Chicago, IL 60611, USA; ishaarganesan2024@u.northwestern.edu

**Keywords:** tumor suppressor, drug development, ubiquitin, p53, MDM2, UBE3A, PTEN, CDK inhibitor, FBXW7, PML

## Abstract

During cancer development, tumor suppressor proteins undergo inactivation due to genetic, epigenetic, or post-translational alterations. While the concept of reactivating tumor suppressor proteins has been appreciated as a potential anti-cancer therapeutic strategy for decades, progress in developing therapies that target tumor suppressor proteins has lagged behind the successful development of therapies targeting oncoproteins such as protein kinases. Here, we review the status and potential of therapeutic targeting against ubiquitin ligases as a promising approach to stabilize and reactivate tumor suppressor proteins.

## 1. Introduction

The loss of functions of tumor suppressor (TS) genes accounts for cancer hallmarks such as impaired DNA damage response, loss of cell-cycle checkpoints, unrestricted proliferation, perturbed apoptosis, metabolic dysregulation, and pro-oncogenic microenvironment [1,2]. Each of the TS genes has been validated primarily by in vivo evidence that shows that either knockdown or knockout of the TS gene is sufficient for spontaneous or enhanced tumorigenesis in animal models, as well as clinical evidence showing that the expression or activity of the protein encoded in the TS gene is downregulated or lost in cancer tissues [3,4]. For instance, *TP53*, one of the most studied TS genes, is mutated or deleted in nearly 50% of all cancers [5], and it has been shown that both homozygous and hemizygous knockout of the *Tp53* gene renders mice highly susceptible to tumorigenesis [6,7,8]. While deletions, promoter methylation, and missense or nonsense mutations in TS genes frequently result in the loss of expression or function of the TS proteins [9,10], downregulation of TS proteins could occur at the translational and post-translational levels [11,12]. The therapeutic potential of reactivating TS proteins in cancers has been appreciated for decades [5,13]; however, clinical applications of the concept have been challenging because of the conceptual and technical difficulties in restoring the expression of TS genes [14]. By contrast, inhibiting oncoproteins, whose ectopic activities drive tumorigenesis, has been more straightforward in nature [15], leading to the successful development of numerous inhibitors against oncogene-encoded kinases and the introduction of 80 or more FDA-approved small-molecule drugs to the clinic [16]. The challenge in developing therapeutic agents to address the loss of TS function is exemplified by the fact that there are few, if any, FDA-approved anti-cancer therapeutic agents that directly target TS proteins for reactivation [14]. Fortunately, this gap in drug development between oncoprotein-targeting and TS-targeting is likely to start diminishing, owing to scientific advances in the field of genetic, epigenetic and biochemical regulation of TS genes, as well as innovations in drug development technologies [15,17,18,19,20]. Among cancer-associated alterations in a TS gene, mutations that result in altered structures of the TS protein, i.e., loss-of-function mutations, would be indications for therapeutic genome editing [21]. Silenced expression of a TS gene by deletion, promoter methylation, or translational defect would be indications for the use of gene therapies to restore expression of the active TS protein. These efforts have been discussed in recent excellent reviews [14,22,23,24]. In this article, we focus on the therapeutic opportunities in the post-translational regulation of TS proteins, especially ubiquitin-dependent degradation.

Recent advances in structural and biological studies on E3 ubiquitin ligases that act as upstream regulators of TS proteins and the development of compounds targeting E3 ligases have provided evidence that the small-molecule-based inhibition of E3 ligases is a feasible and promising strategy to therapeutically restore the expression of TS proteins in cancer tissues, as far as the TS proteins retain wild-type structures [25,26]. Table 1 shows E3 ligases known to function as upstream regulators of representative TS proteins. Small-molecule-based interference with the interaction between an E3 and its substrate TS protein or inhibition of the E3 catalytic activity would prevent the TS protein from undergoing ubiquitin-dependent degradation, leading to enhanced stability and upregulation of the TS protein [26]. This approach is expected to pave a promising avenue for the much-needed development of TS-targeting therapies. Indeed, the past decade has seen several small-molecule E3 inhibitors (e.g., MDM2 (Mouse double minute 2)) introduced into clinical trials [27]. Furthermore, recent clinical studies have shown that E3 inhibitors could have sufficient therapeutic efficacies when used in specific genetic or biological contexts and in combination with other treatments [27,28,29,30]. In this article, we will first highlight the E3s upstream of bona fide TS proteins that have been or can be targeted for inhibition to stabilize their respective TS substrates (Table 1), then review the current status of E3 inhibitors under the developmental pipeline and finally discuss the promise and precautions of E3-targeting therapies in precision medicine.

## 2. p53: “The Guardian of the Genome”

p53 is a bona fide TS protein which plays a variety of tumor-suppressive roles including cell-cycle checkpoint functions, apoptosis induction, mitochondrial regulation, metabolic roles, and transcriptional control. Implications of p53 inactivation in tumorigenesis have been well described in these excellent reviews [8,80,81,82,83]. *TP53*, the gene encoding the p53 protein, has been found to be mutated or deleted in nearly 50% of cancers [3,84]; however, the frequency of *TP53* mutations varies heavily among different cancer types, and some malignancies such as cervical cancer, renal clear cell carcinoma and leukemia have less than 10% *TP53* mutational frequency (Figure 1). Cancers that retain wild-type *TP53* offer salient targets for therapeutic inhibition of upstream E3 ligases, which is expected to render positive therapeutic outcomes with stabilization and reactivation of p53 [85]. Restoring the expression of wild-type p53 in cancer cells would result in molecular changes with a particular signature, including the upregulation of genes transactivated by p53, such as p21, MIC-1, PUMA, BAX, BID, BAK, and NOXA, which can be used as therapeutic biomarkers of successful p53 stabilization [81,86,87]. Histological markers of apoptosis in tumor tissues [88,89,90,91] could be enhanced upon p53 stabilization. Additionally, activation of wild-type p53 has been shown to upregulate anti-angiogenic factors (TSP1, BAI1) and downregulate pro-angiogenic factors (VEGF, bFGF, COX-2) [86]. A number of E3s have been demonstrated to ubiquitinate p53 (Table 1). Among them, we will focus on the broadly implicated p53-MDM2 axis and the human papillomavirus (HPV)-associated E6-UBE3A-p53 axis as prototypes that exemplify the potential of therapeutic E3 targeting to reactivate TS proteins.

### 2.1. Mdm2 Inhibition Stabilizes Wild-Type P53 in Cancers

The RING-family E3 ligase MDM2, whose human ortholog is also known as HDM2, is thought to be the master regulator of p53 that determines basal turnover of this TS protein [93] (Figure 2A). The biological significance of the MDM2-p53 axis is highlighted by the studies showing that *Mdm2*-null mice died early in development, but this embryonic lethality was abolished by co-disruption of the *Tp53* gene [36,94]. MDM2 appears to inhibit the tumor-suppressive function of p53 in multiple steps. MDM2 binds to p53 at its DNA-binding domain which directly inhibits p53-mediated transactivation [95]. MDM2 then mediates polyubiquitination of p53 to promote degradation by nuclear proteasomes [96]. When the nuclear level of MDM2 is low, MDM2 may monoubiquitinate p53, leading to the transport of p53 to the cytoplasm and suppressed transactivation [96]. The control of the E3 activity and nuclear localization of MDM2 depends on its association with MDMX/MDM4, as reviewed by these excellent review articles [97,98]. While p53 phosphorylation is not a prerequisite for MDM2-mediated ubiquitination, phosphorylation of MDM2 itself by the checkpoint kinase ATM inhibits MDM2 activity, promoting p53 stabilization in response to DNA damage [99]. Furthermore, the MDM2-p53 axis is fine-tuned by p53 expression levels, since MDM2 is a target of p53-dependent transactivation, forming a feedback loop to prevent the cell from accumulating excessive p53 [100]. Thus, overexpression of MDM2 observed in various cancers perturbs the basal autoregulatory mechanism of the p53-MDM2 axis, promoting oncogenesis.

### 2.2. Mdm2 Inhibition Causes P53 Stabilization and Suppresses Tumorigenesis in Wild-Type P53 Cancers

It has been demonstrated how MDM2 inhibition can cause tumor suppression both in in vitro and in vivo models of various cancers [28,86,101]. Since direct substrate interaction is crucial for the ubiquitin ligase activity of RING-E3s [102], a feasible therapeutic approach is to interfere with the protein–protein interaction (PPI) between p53 and MDM2. The physical interaction between p53 and MDM2 is driven by key residues of p53 (Phe19, Trp23, Leu26) in its N-terminal DNA-binding domain burying into a hydrophobic cleft in the N-terminal domain of MDM2 [103,104]. In 2004, the first small-molecule inhibitor for the MDM2-p53 binding pocket was developed, called Nutlin-3a [105]. This compound was part of a series of cis-imidazoline derivatives called Nutlins that were designed to mimic the interaction of the three residues of p53 with the MDM2 cleft. Nutlin-3a showed robust inhibition of p53-MDM2 binding in vitro and tumor suppression with p53 upregulation in vivo, and served as a prototype in developing next-generation MDM2 inhibitors that are currently being tested in clinical trials [85]. In addition, there are other small-molecule MDM2-p53 inhibitors outside the Nutlin class such as spiro-oxindole compounds [106], piperidin-2-one scaffold-containing compounds [107,108], and dihydroisoquinolinone-based compounds [109], which mimic the critical residues of p53 for binding MDM2 with high affinity. The preclinical development of these inhibitors successfully underwent multiple prerequisite steps, such as structural assessment of the interface between the E3 (MDM2) and the substrate TS protein (p53), identification of small molecules that interfere with the interaction by high-throughput screens or molecular-modeling-based virtual screens, subsequent biochemical and pharmacological validation, and further optimization in in vivo models [85,110,111].

Furthermore, compounds that disrupt the E3 ligase activity of MDM2 mediated by its RING domain have also been shown to promote p53 stabilization [112,113]. While such enzymatic inhibitors would not specifically target p53-MDM2 binding, MDM2 RING inhibitors offer an orthogonal approach to stabilize p53 by blocking its ubiquitination. The in vivo observation that partial loss of MDM2 caused by a hypomorphic *Mdm2* allele can still exert tumor suppression in mice without the premature aging phenotype linked to excess p53 activation provides additional rationale for the development of MDM2 RING inhibitors which may render proper therapeutic windows to inhibit the MDM2-p53 axis [114]. Although the development of RING inhibitors has been generally challenging due to technical issues in crystalizing RING domains [115], several MDM2 RING inhibitors have been identified with promising therapeutic potential. 5-deazaflavin analogs, hinokiflavone, pyridoacridine alkaloids, and sempervirine have been shown to bind to the MDM2 RING domain and inhibit its E3 ligase activity. These compounds can stabilize p53 in cancer cells and promote tumor-suppressive and pro-apoptotic outcomes [116,117,118]. At this moment, these MDM2 RING inhibitors remain in the preclinical stage. It is worth noting that small-molecule inhibitors targeting MDMX-p53 and MDMX-MDM2 interactions have been developed, both of which stabilize p53 through mechanisms distinct from those of the MDM2 inhibitors [119,120]. The progress in developing MDM2 RING and MDM2/MDMX inhibitors underscores the importance of exploring diverse modalities to achieve the long-sought therapeutic objective of TS reactivation.

### 2.3. Mdm2 Inhibitors—The Latest Progress

In the past twenty years, many clinical studies have been carried out to evaluate the efficacy of MDM2 inhibitors in stabilizing p53 and respective therapeutic outcomes. While there are no FDA-approved MDM2 inhibitors on the market yet, one (KT-253) has received an FDA orphan drug designation for AML and a few others are being evaluated in Phase III studies for the treatment of *TP53* wild-type cancer [85]. MDM2 inhibitors are the only category of E3 inhibitor drugs that have hitherto reached the stage of clinical trials. An excellent review published in 2024 by Wang et al. comprehensively summarizes the status of clinical trials with MDM2 inhibitors across various stages, as well as the outcomes of closed clinical trials [85]. Here, we will briefly discuss the inhibitors farthest along in clinical trials.

The new-generation compounds targeting the MDM2-p53 axis such as KT-253, Brigimadlin, Milademetan, and Navtemadlin are drivers of a new era in TS-stabilizing therapeutics. Importantly, the upregulation of biomarkers associated with p53 activation, as observed in KT-253 trials, demonstrated the significance of molecular indicators in assessing treatment success (NCT05775406) [121]. The advance of Brigimadlin to Phase III trials underscores its potential in addressing rare cancers like dedifferentiated liposarcoma, while its application to *TP53*-wild-type tumors with MDM2 overexpression points to broader therapeutic horizons (NCT05218499) [122,123]. Milademetan did not meet its primary endpoint in Phase III trials but is still being evaluated by further clinical trials (NCT05012397; NCT04979442) [124,125]. Likewise, Navtemadlin has shown positive therapeutic potential in myelofibrosis, warranting the development of synergistic therapeutics (NCT03662126; NCT04485260).

The clinical development of MDM2 inhibitors faces several challenges. First, the risk of drug resistance due to *TP53* mutations or the upregulation of alternative oncogenic pathways poses significant hurdles. Second, patient selection is critical, as MDM2 inhibitors are most effective in cancers that retain wild-type p53 and overexpress MDM2. The success of MDM2 inhibitors in the clinic will inevitably require a comprehensive profile of reliable biomarkers to predict patient responses. Third, adverse effects, particularly hematological and gastrointestinal toxicities, have been observed across many MDM2 inhibitors [85]. Strategies to mitigate these toxicities necessitate further optimization of dosing regimens and exploration of novel targeted delivery methods to minimize systemic exposure. The ongoing trials with KT-253, Brigimadlin, Milademetan and Navtemadlin will provide critical insights about adverse effects.

### 2.4. Synergistic Therapies—The Future of MDM2 Inhibitor Therapies

The progress in the development and application of MDM2 inhibitors provides critical implications as a prototype for developing inhibitors of other E3s -upstream TS proteins. While MDM2 inhibitors have been tested in the context of monotherapies, currently available data suggest that E3-targeting therapies should be integrated into personalized combinational therapies, particularly due to the ever-present obstacle of drug resistance. Resistance to MDM2 inhibitors could arise from genetic alterations, including *TP53* mutations and adaptations in other oncogenic pathways, analogous to the mechanisms of resistance observed with oncogenic kinase inhibitors. For example, cancers harboring co-occurring mutations in pathways such as PI3K/AKT or RAS/MAPK may bypass the effects of p53 stabilization by activating alternative survival mechanisms [126,127]. MDM2 inhibitor resistance has been characterized by upregulation of anti-apoptotic proteins such as BCL-2 and MCL-1, which neutralize the pro-apoptotic effects of p53 reactivation. Such resistance underlines the common challenge in the use of monotherapies in treating cancers, particularly tumors with high intratumoral plasticity. To address these mechanisms, MDM2 inhibitors have been combined with agents targeting these compensatory pathways (i.e., PI3K/AKT, MAPK), showing better therapeutic outcomes in preclinical models [128]. For example, in neuroblastoma models, combining MDM2 inhibitors with MYC inhibitors delayed tumor growth and prolonged survival, compared to MDM2 inhibitor monotherapy [86]. Similarly, in thyroid and colon cancer models, combining MDM2 inhibitors with MEK inhibitors effectively reduced cell proliferation, increased apoptosis, and delayed resistance development [28,85,129,130,131,132,133]. Another key contributor to therapy resistance is the tumor microenvironment (TME). Stabilizing p53 in cancer cells may not sufficiently alter the TME to facilitate long-term tumor control; for example, the hypoxic regions of tumors often harbor cells that are resistant to apoptosis, even when p53 is stabilized [134]. Adjacent to TME-driven resistance, the senescence-associated secretory phenotype (SASP) induced by p53 reactivation may facilitate development of various forms of drug resistance, as demonstrated in breast cancer models [135]. To address these additional resistance mechanisms, MDM2 inhibitors have been used in conjunction with immune checkpoint inhibitors (ICIs), such as anti-PD-1 and anti-PD-L1 therapies, to enhance anti-tumor immunity. Preclinical models combining MDM2 inhibitors with ICIs have shown that p53 stabilization may enhance recognition of cancer cells by the immune system, resulting in synergistic tumor regression [136]. These data highlight the importance of strategic uses of MDM2 inhibitors in dual-modality approaches [137]. Another benefit of combinational therapies involving MDM2 inhibitors is better control over the dose-limiting toxicities associated with MDM2 inhibitor monotherapies, particularly hematological effects such as neutropenia and thrombocytopenia, as elucidated across clinical trials [36]. These toxicities stem from p53 activation in highly proliferative normal tissues, including bone marrow, where high turnover rates make cells susceptible to p53-induced apoptosis [77,78]. Combinational therapies may allow dose adjustments of individual agents, potentially reducing the toxicity burden while maintaining efficacy [27,85]. For instance, modest doses of AMG232, a Nutlin derivative, paired with lower doses of radiation, were shown to promote therapeutic outcomes in preclinical cancer models, without exacerbating adverse effects [133]. In sum, these findings emphasize the importance of employing combinational strategies with future MDM2 inhibitor therapeutics to overcome resistance and achieve durable therapeutic outcomes.

### 2.5. UBE3A/E6AP

#### 2.5.1. Background: UBE3A Biology and the HPV E6-UBE3A Axis

The discovery of UBE3A as a critical player in human papillomavirus (HPV)-associated oncogenesis has provided insights into the mechanisms of p53 degradation [138] (Figure 2B). UBE3A, also known as E6-associated protein (E6AP), is a HECT-type E3 ubiquitin ligase with roles in both physiological development and pathological processes. Under normal conditions, UBE3A mediates the ubiquitination of substrates involved in cell signaling and neuronal development, and *UBE3A* loss of function in brain causes Angelman syndrome, a neurodevelopmental disorder [138]. In the context of HPV infection, UBE3A is hijacked by the E6 viral oncoprotein to target p53 for proteasomal degradation. HPV-associated cancers, which include cervical, oropharyngeal, and anogenital cancers, are predominantly caused by high-risk HPV types such as HPV16 and HPV18 [139]. Without E6, UBE3A is inefficient to ubiquitinate p53, highlighting the impact of HPV infection on the substrate specificity of UBE3A [140]. Mechanistically, E6 binding to UBE3A induces a conformational change that activates its E3 ligase function. The E6-UBE3A complex recruits p53, facilitating its polyubiquitination and subsequent degradation via the 26S proteasome [141,142]. The E6-UBE3A-p53 ternary complex formation is sufficient for p53 ubiquitination, without involving phosphorylation of the E3 or the substrate. Akin to MDM2-mediated negative regulation of p53, E6-UBE3A disables the tumor-suppressive functions of p53, leading to unrestricted proliferation, impaired apoptosis and genomic instability toward the development of HPV-associated cancers [143]. The E6-UBE3A-p53 interaction is unique in HPV-infected epithelial cells and distinct from the physiological control of the basal turnover of p53 by MDM2, making this pathway a specific and attractive target for therapeutic intervention.

#### 2.5.2. Development and Future Directions of Therapeutics Targeting E6-UBE3A

Therapeutic development targeting the E6-UBE3A-p53 axis has primarily focused on disrupting PPIs within the ternary complex [144]. Several PPI inhibitors have demonstrated preclinical efficacy by stabilizing p53 and inducing tumor-suppressive effects in HPV-infected models. For example, Amici et al. (2016) developed an intracellular antibody (“intrabody”) targeting the E6 protein, which impaired E6 binding to UBE3A and resulted in reduced tumor growth in HPV16-positive cancer models [145]. Similarly, small molecules targeting the E6-p53 interaction, such as the covalent peptide inhibitors termed “reactides”, specifically disrupted the LXXLL binding motif of UBE3A, effectively blocking E6-dependent activation of this E3 [146]. These inhibitors restored p53 stability in HPV-positive cervical cancer cells, with biomarkers such as elevated p21 and PUMA levels indicating successful p53 reactivation. Additional efforts to block the E6-UBE3A interaction have identified more small molecules [147,148,149]. These compounds demonstrated efficacy in suppressing tumor growth in HPV-associated cancer models with concomitant increases in p53-dependent transactivation and decreases in E6-UBE3A complex formation, further validating E6-UBE3A as a viable therapeutic target in HPV-driven malignancies. While these studies shared the overarching goal of antagonizing the E6-UBE3A-mediated degradation of p53, they varied in methodology; in essence, they targeted different aspects of the E6-UBE3A-p53 interaction (e.g., direct inactivation of E6, inhibition of E6 stabilizing factors such as GRP78, and interference with the formation of the E6-UBE3A-p53 complex). As discussed in Section 2.2, identifying differentially targeting inhibitors for an E3-TS axis is important for the development of next-generation therapeutics, to stratify therapies capable of targeting multiple vulnerabilities within the oncogenic driver function of E6-UBE3A-p53. Furthermore, targeting the catalytic domain of UBE3A has emerged as a strategy alternative to the PPI inhibitors, since high-resolution structural studies have provided more insights into the ubiquitin transfer mechanism of UBE3A, paving the way to rational drug design [141]. Small-molecule-based inhibition of the E3 activity of UBE3A would perturb not only E6-dependent ubiquitination of p53 but also other physiological substrates, which may elicit a concern about adverse effects when UBE3A enzymatic inhibitors are used to treat HPV-associated cancers. However, induced disruption of the *Ube3a* gene in postnatal mice does not cause any appreciable phenotype [150], suggesting that on-target adverse effects of UBE3A inhibitors would be minimal when used in adult cancer patients. Thus, developing enzymatic inhibitors of UBE3A is a viable option for treating HPV-associated cancers, given the general challenges associated with the development of potent and selective PPI inhibitors.

Future directions for the development of E6-UBE3A inhibition therapeutics include the exploration of combinational therapies to enhance the efficacy of E6-UBE3A inhibitors. Combining these inhibitors with DNA-damaging agents or ICIs may produce synergistic effects by simultaneously restoring p53 function and sensitizing tumors to immune-mediated clearance [140]; Li et al. showed the role of LOC644646 (a long non-coding RNA) and ZNF143 in modulating the E6-UBE3A interaction for p53 degradation, in the context of cisplatin resistance [151]. Furthermore, therapeutic HPV vaccines under the developmental pipeline could exert synergistic therapeutic effects with E6-UBE3A inhibition in HPV-associated cancers [152]. Leveraging biomarkers such as HPV genotypes and the expression levels of E6, p53 and representative p53-transactived genes in tumor tissues could enable the stratification of patients most likely to benefit from these therapies. Thus, the E6-UBE3A-p53 axis represents a highly specific therapeutic target for HPV-associated cancers. The continued development of small-molecule inhibitors, supported by innovative drug delivery- and biomarker-driven approaches, holds the promise of improving outcomes for patients with these malignancies; the refinement of these strategies is expected to provide precision-based therapies against HPV-associated cancers.

### 2.6. Other E3s Upstream of p53

MDM2 acts as a master regulator of the basal turnover of p53, and UBE3A is the key player in accelerated degradation of p53 in HPV-associated cancers, as described above. Besides these E3s, there are several other E3 ubiquitin ligases known to mediate p53 polyubiquitination for subsequent proteasomal degradation (listed in Table 1). For instance, HUWE1 (HECT, UBA and WWE domain-containing E3 ubiquitin protein ligase 1) is frequently overexpressed in lung cancer, which correlates with poor prognosis. Inhibiting HUWE1 can suppress tumorigenicity in in vivo lung cancer models, with concomitant stabilization of p53 [34,153]. COP1 (Constitutive Photomorphogenic 1) is another E3 ligase that mediates p53 polyubiquitination and has been found to be overexpressed in breast and ovarian cancer, in association with decreased p53 expression [32,154]. PIRH2 (p53-induced protein with a RING-H2 domain, also known as Rchy1), like MDM2, participates in an autoregulatory feedback loop with p53 [31]. The PIRH2 protein is overexpressed in lung cancer, specifically non-small-cell carcinomas, correlating with decreased expression of p53 due to accelerated degradation [155,156,157,158]. Thus, the cancer-relevant properties of these E3s provide a rationale for developing inhibitors of each of these enzymes as anti-cancer therapeutics. Exemplified by the prevalence of PPI inhibitors targeting the MDM2-p53 interaction in clinical trials, structural studies have been promoting the development of small molecules to target specific E3-p53 interactions for therapeutic inhibition [159,160,161]. To develop enzymatic inhibitors of E3s upstream of p53, physiological substrates of each E3, besides p53, should be defined comprehensively to ensure minimal or tolerable adverse effects of therapeutic targeting. Some of the p53-regulating E3s listed in Table 1 have been shown to mediate the degradation of other substrates including oncoproteins, although the studies did not have proteome-wide scopes. For example, COP1 may also play a tumor-suppressive role as it mediates the degradation of the oncoprotein c-Jun, which makes the therapeutic potential of COP1 inhibition uncertain at this time [162]. Physiological substrates of MDM2 and UBE3A have been extensively studied using various proteomic techniques such as ubiquitome profiling by quantitative TMT proteomics combined with di-Gly-dependent enrichment of ubiquitinated peptides [163] and the orthogonal ubiquitin transfer (OUT) as a catalysis-based ubiquitin-tracking approach [164]. Thus, similar studies should be performed to profile the substrates of HUWE1, COP1, PIRH2, and the other E3s upstream of p53 in a proteome-wide manner, so that the potential of these E3s as anti-cancer therapeutic targets could be assessed and established.

## 3. PTEN

PTEN (phosphatase and tensin homolog) is a TS protein whose function is primarily attributed to its inhibition of the PI3K/AKT signaling pathway, suppressing cell proliferation and survival [165,166,167]. Beyond this canonical role, PTEN has AKT-independent phosphatase functions [168,169]. Although the *PTEN* gene is not altered as frequently as *TP53* across cancers (Figure 3), even subtle reduction in PTEN protein levels can disrupt tissue homeostasis and induce tumor initiation and progression [170,171,172]. Additionally, most cancers (except for glioblastoma) retain at least one wild-type *PTEN* allele [173,174]; thus, developing E3-targeting therapies to stabilize the wild-type PTEN protein will be of remarkable value. The ubiquitination status of PTEN plays a critical role in regulating the stability and activity of PTEN. Polyubiquitination typically targets PTEN for proteasomal degradation, whereas monoubiquitination is associated with its nuclear translocation, where it carries out its regulatory roles in genomic stability [175]. Thus, targeting the E3s upstream of PTEN may enable precise therapeutic control of PTEN reactivation, based on what type of ubiquitination the targeted E3 catalyzes. A recent review highlights the intricate interplay between PTEN structure and its various forms of ubiquitination [175]. While numerous E3 ubiquitin ligases have been implicated in PTEN regulation, this review will discuss only E3 ligases with established oncogenic roles. Other E3 ligases upstream of PTEN appear to play context-dependent dichotomous roles in controlling oncoproteins and TS proteins and require further profiling to validate their therapeutic potential.

### 3.1. NEDD4-1-PTEN Axis

NEDD4-1 (neural precursor cell-expressed developmentally downregulated protein 4-1) was the first E3 ligase identified to mediate PTEN ubiquitination and degradation [48]. The interaction between NEDD4-1 and PTEN is mechanistically analogous to the MDM2-p53 axis. Both NEDD4-1 and MDM2 function as the primary regulators of protein turnover of PTEN and p53, respectively [48]. Consequently, NEDD4-1 is a compelling therapeutic target to restore PTEN stability and activity in cancers. The NEDD4-1-PTEN axis is implicated in the progression of non-small-cell lung cancer, breast cancer, glioblastoma, prostate cancer, pituitary adenoma, pancreatic ductal adenocarcinoma, bladder cancer, melanoma, and gastric cancer. NEDD4-1 overexpression inversely correlates with PTEN protein levels in these cancers [48]. Furthermore, NEDD4-1 knockdown or knockout results in PTEN reactivation with associated biomarkers such as reduced AKT phosphorylation, decreased IGF-1 signaling, and inhibited tumor growth with enhanced cell death [176].

### 3.2. Therapeutic Potential of Targeting NEDD4-1

Preclinical studies have demonstrated that inhibiting NEDD4-1 can stabilize PTEN and suppress the progression of tumors exhibiting NEDD4-1 overexpression. Small molecules such as Heclin (a non-specific HECT E3 inhibitor) [177] and Indole-3-carbinol (I3C; inhibitor against NEDD4-1 and WWP1 catalytic HECT domains) [178,179] are capable of stabilizing PTEN, reducing Akt phosphorylation, and impairing cell viability in models of gastric cancer and melanoma [48]. However, significant difficulties exist in the development of NEDD4-1 inhibitors. Achieving specificity remains a primary hurdle, as NEDD4-1 shares structural similarities with other HECT-domain E3 ligases, challenging the development of enzymatic inhibitors [180]. Although the analogy between the NEDD4-1-PTEN axis and the MDM2-p53 axis provides a conceptual framework for therapeutic development, there are notable distinctions that complicate the design of targeted interventions against NEDD4-1 [181,182]. MDM2 regulates p53 as a predominant substrate, whereas NEDD4-1 has a broader spectrum of substrates, which paradoxically includes MDM2 [183]. Future directions should prioritize the development of selective NEDD4-1 inhibitors, such as disruptors of E3–substrate interactions or allosteric inhibitors. Additionally, combining NEDD4-1 inhibitors with PI3K/AKT/mTOR pathway inhibitors or other chemotherapeutic agents may enhance efficacy and overcome resistance. For instance, I3C can sensitize glioblastoma cells resistant to the alkylating agent temozolomide, exerting synergistic effects [184]. Biomarker identification is another critical step to select cancer types reliant on NEDD4-1-mediated PTEN degradation, enabling better patient stratification for optimal therapeutic outcomes.

### 3.3. Other E3 Ligases Mediating PTEN Ubiquitination

Several other E3 ubiquitin ligases are also known to mediate PTEN polyubiquitination and degradation in cancer type-specific ways. For example, WWP1 (WW domain-containing E3 ubiquitin protein ligase 1) is frequently overexpressed in papillary thyroid cancer [185] and breast [186,187], prostate [188], and gastric cancers [189]. Gain of WWP1 function is implicated in the genetic landscape of a subset of inherited cancer patients characterized by PTEN loss [47,190] and appears to drive oncogenesis in both PTEN-dependent and independent mechanisms. Recent studies have shown that I3C-mediated inhibition of WWP1 can restore PTEN stability and suppress tumor progression in MYC-driven cancer models [178], implying how therapeutic targeting of the WWP1-PTEN axis could be effective against a broad spectrum of cancers driven by the MYC oncogene. Similarly, the WWP2 (WW domain-containing E3 ubiquitin protein ligase 2)-PTEN axis [191] has been implicated in prostate, endometrial, hepatocellular, oral squamous cell cancers, and glioma [46,192,193,194,195], where WWP2 overexpression correlates with aggressive tumor phenotypes and poor prognosis. Notably, WWP2 promotes degradation of phosphorylated PTEN, in contrast to NEDD4-1, which preferentially ubiquitinates unphosphorylated PTEN [196]. Therefore, elucidating the specific phosphorylation profiles for PTEN in cancers will be important in determining which E3 should be targeted. While therapeutic inhibitors for WWP1 and WWP2 are still in early stages of development, targeting these E3s is expected to improve therapeutic outcomes through PTEN stabilization [175,197]. XIAP (X-linked inhibitor of apoptosis protein) is another E3 ligase upstream of PTEN, which can polyubiquitinate PTEN for proteasomal degradation or monoubiquitinate PTEN for nuclear localization [45]. The XIAP-PTEN axis has been implicated in the malignant phenotypes of pancreatic and ovarian cancers, as well as those in glioblastoma [181,198,199]. Primarily known for its anti-apoptotic role by mediating the degradation of pro-apoptotic factors, XIAP has been extensively studied as a therapeutic target [200]. While clinical trials are ongoing to evaluate small-molecule inhibitors of the inhibitor of apoptosis protein (IAP)-family proteins including XIAP, such as SMAC (Second Mitochondrial-derived Activator of Caspases), these compounds do not directly inhibit the E3 activity of XIAP [201,202]. The pro-oncogenic roles of WWP1, WWP2, and XIAP in regulating PTEN stability highlight their potential as therapeutic targets. However, the context-dependent functions of these E3 ligases present challenges for drug development. As discussed above, substrates beyond PTEN must be comprehensively profiled to minimize adverse effects. Thus, proteome-wide approaches such as ubiquitome profiling or catalytic ubiquitin tracking like OUT are essential to assess the therapeutic potential of these E3s, similar to the substrate profiling for MDM2 and UBE3A. As future precision medicines will inevitably require targeting multiple mechanisms of oncogenesis, the use of E3 ligase inhibitors for PTEN stabilization in conjunction with other therapies targeting upstream (e.g., MYC-WWP1 axis) or downstream pathways (e.g., PI3K/AKT/mTOR) must be investigated. Supported by the positive outcomes of combinational therapies involving MDM2 inhibitors, therapies targeting E3s upstream of PTEN may offer synergistic benefits in overcoming drug resistance and improving treatment outcomes in cancers with reduced PTEN expression.

## 4. p27^Kip1^

p27, the cyclin-dependent kinase inhibitor (CKI) encoded in the *CDKN1B* gene, is a key regulator of the G1-S progression in the cell cycle, associating with cyclin D- and cyclin E-dependent kinases [203]. As a TS protein, its degradation is controlled by multiple E3 ligases to ensure proper cell-cycle control. While *CDKN1B* mutations are rare, decreased levels of p27, often mediated by aberrant activity of E3 ligases, are likely to contribute to tumorigenesis and poor prognosis in various cancers [204,205]. However, it is noteworthy that in some cellular contexts, p27 may function as an oncoprotein especially in the cytoplasm, as shown in osteosarcoma [206]. The primary E3-mediating ubiquitination of phosphorylated p27 in the nucleus at the G1/S boundary is the Skp1–Cullin–F-box (SCF) complex including SKP2 as an F box protein, hereon referred to as SKP2 [59,207]. SKP2 plays a dominant role in temporal regulation of p27 in the cell cycle, mediating p27 ubiquitination upon CDK2-dependent phosphorylation of this CKI at the threonine-187 residue [59]. SKP2-mediated degradation of p27 has been regarded as an oncogenic driver [208] and the SKP2-PTEN axis is implicated in breast cancer, lung cancer, prostate cancer, colorectal cancer, gastric cancer, and prostate cancer [209]. While there are no SKP2 inhibitors currently under clinical trials, preclinical assessment of SKP2 inhibitors using in vitro and in vivo cancer models [208,210,211,212] supports their potential for therapeutic reactivation of p27 and induction of cell-cycle arrest in cancers. SKP2 inhibitors may be useful to treat RB-deficient cancers for which CDK4 inhibitors are ineffective [213,214] and also show efficacy in resensitizing drug-resistant cancers [215,216] and synergistic effects with other drugs [215,217]. Another recently discovered E3 upstream of p27 is RNF6, which mediates phosphorylation-independent ubiquitination and degradation of nuclear p27 in the early G0/G1 phase. RNF6 has been postulated to be a promising target for therapeutic inhibition to promote apoptosis in in vitro models of prostate cancer, leukemia, and multiple myeloma [57,218]. Development of RNF6 inhibitors has already begun, prompted by other oncogenic roles of this E3 ligase in mediating activation of the STAT3 and Wnt/B-catenin pathway [219]. In addition to RNF6, there are other E3 ligases mediating p27 turnover and implicated as oncoproteins (Table 1). Further characterization of these E3 ligases regulating p27 may uncover novel therapeutic opportunities in cancers with low p27 expression.

## 5. FBXW7

FBXW7/FBW7 is a TS protein that forms an SCF complex with well-characterized E3 ligase activity to ubiquitinate and degrade various oncoproteins including c-Myc, cyclin E, c-Fos, Notch-1, and mTOR [220]. The loss of *FBXW7* function, often due to mutations of the gene, is an oncogenic driver in colorectal and breast cancers and hematological malignancies [221]. FBXO45 has been shown to function as an upstream E3 ligase mediating FBXW7 ubiquitination and degradation [60]. FBXO45 itself is a known oncoprotein and highly expressed in cancers such as lung, gastric, and pancreatic cancers, with its elevated levels often correlating with poor survival [222,223]. The FBXO45-FBXW7 axis appears to be involved in not only tumor progression but also drug resistance. Mechanistically, FBXO45, in complex with MYCBP2/PAM, ubiquitinates and degrades FBXW7 during spindle toxin-induced mitotic arrest, reducing cellular sensitivity to nocodazole and Taxol and inhibiting mitotic cell death [60]. Interestingly, FBXO45 can indirectly upregulate Polo-like kinase 2 (PLK-2), which, in turn, phosphorylates FBXW7 to facilitate its ubiquitination and degradation [224]. Thus, inhibiting FBXO45 would disrupt PLK-2-driven autoubiquitination of FBXW7, further stabilize FBXW7 and enhance mitotic cell death induced by the spindle toxins. Further studies are warranted to develop a robust profile of other substrates of FBXO45, as well as to validate the FBXO45-FBXW7 axis as a feasible therapeutic target.

## 6. PML

Promyelocytic leukemia (PML) protein is a TS protein that plays diverse roles in cellular senescence, apoptosis, angiogenesis, cell migration, and the maintenance of genomic stability [225]. While the *PML* gene is involved in chromosomal translocation typically in acute promyelocytic leukemia, loss of PML protein expression rather than *PML* mutations or deletions has been demonstrated in various cancers [226]. Localized in the PML nuclear bodies (PML-NBs), PML serves as a scaffold for various proteins involved in tumor suppression including p53. Quite a few E3s have been identified to mediate the proteasomal degradation of PML, including KLHL20, UHRF1, UBE3A and RNF4, which may drive the development of cancers at least partially through PML downregulation (Table 1). Thus, there are opportunities for the development of PML stabilization therapies by inhibiting the E3-PML interactions. Under hypoxic conditions, KLHL20 promotes the ubiquitination and degradation of PML [227,228]. In the hypoxic tumor microenvironment, KLHL20 at elevated levels targets PML for constitutive degradation, reducing the number of PML-NBs and impairing tumor-suppressive functions. KLHL20-mediated PML ubiquitination depends on phosphorylation of PML by CDK1 or CDK2 and subsequent prolyl isomerization by PIN1 [69]. PML downregulation mediated by KLHL20 has been linked to the progression of prostate and colon cancers. Potential biomarkers of cancers with enhanced KLHL20 activity to degrade PML include increased HIF-1α and Pin1 levels [227]. UHRF1 is a nuclear RING finger E3 ligase that mediates the proteasomal degradation of PML [68]. UHRF1 is overexpressed in several cancer types, including lung and breast cancers [229]. While URHF1 is a pro-oncogenic protein functioning as a transcriptional repressor of key TS genes such as *CDKNA1, CDKNB1, MLH1*, and *RB1* [68,229], its E3-activity associated role in tumorigenesis remains elusive, and more studies are necessary to validate the UHRF1-PML axis as a therapeutic target. Another E3 as a potential therapeutic target upstream of PML is UBE3A [67]. In *Ube3a*-null mice, elevated levels of PML and PML-NBs were observed across multiple tissues, with elevated sensitivity to apoptosis upon genotoxic stress. Furthermore, *UBE3A* haploinsufficiency in an MYC-driven B-cell lymphomagenesis model increased PML levels, induced cellular senescence, and suppressed tumor progression, which is consistent with elevated levels of UBE3A and decreased levels of PML in primary B-cell lymphoma samples [230]. Similar oncogenic roles of the UBE3A-PML axis were observed in prostate cancer models [231]. These findings support the therapeutic potential of UBE3A enzymatic inhibitors to treat non-viral cancers.

## 7. Conclusions

Since the early 2000s, targeting E3 ubiquitin ligases that control the stability of their substrate TS proteins has been evaluated as an emerging strategy to treat a variety of cancers [232]. In 2005, Garber wrote an article describing the “stall” in E3 inhibitor development [233]; nearly two decades later, we have yet to see the first bona fide FDA-approved E3 inhibitor. A major difficulty in developing therapies to target the E3-TS axes is due to the lack of well-defined catalytic domains in most E3s. For instance, RING E3s which form a vast majority of ubiquitin ligases in the proteome do not have catalytic cysteine residues, unlike HECT E3s, and mediate the ubiquitination of their substrates primarily through its PPIs with substrates and E2-ubiquitin complexes [234]. Thus, the development of PPI inhibitors is feasible only when a druggable E3-TS binding interface is present and well defined structurally. Moreover, two other major issues that are delaying the introduction of E3 inhibitors to the clinic are low efficacy and dose-limiting toxicities. Indeed, these are the primary challenges next-generation MDM2 inhibitors have faced in clinical trials. Nonetheless, technological advances such as high-resolution crystal structure studies, novel in silico molecular modeling techniques and methodologies uncovering E3–substrate profiles have reinvigorated the development of E3-targeting therapeutics. Leading the way, the prototypical development of MDM2 inhibitors and UBE3A inhibitors has provided a foundation for stratifying the development of more E3-targeting therapies for TS reactivation. Such therapies depend on the post-translational upregulation of TS proteins and require the retention of wild-type structures of TS proteins in cancer cells. E3 ligases for therapeutic targeting should be validated by experimental and clinical evidence indicating their unambiguous oncogenic functions and dominant roles in determining the expression levels of their substrate TS proteins in particular cellular contexts, e.g., basal turnover and constitutive degradation triggered by viral infection, E3 overexpression or other stimuli. As the future of drug development lies in precision medicine integrated with molecular signatures of individual tumors, it is crucial that we apply orthogonal methodologies to examine the proteome-wide impact of therapeutic TS targeting in various cellular contexts and genetic backgrounds to develop highly potent and selective E3 inhibitors. The optimal treatment of a cancer patient will need to address the unique genomic and proteomic landscape in that respective cancer. It is likely that such individualized medicine will involve combinations of both oncogene inhibitors and TS-reactivating therapies based on the patients’ specific profiles.

## Figures and Tables

**Figure 1 cancers-17-00626-f001:**
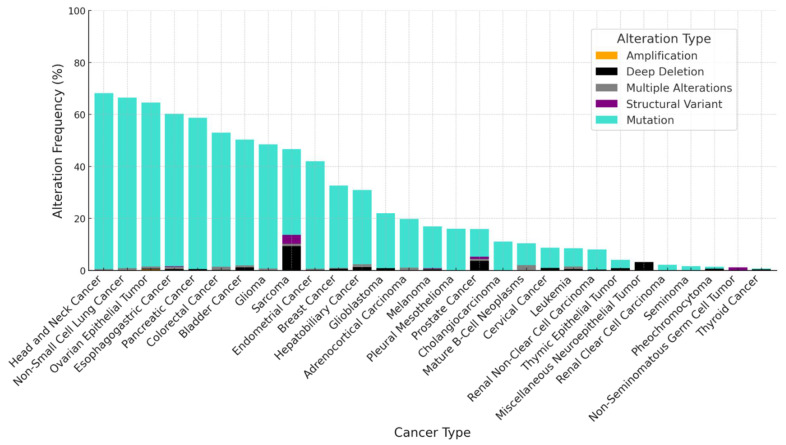
Frequency of genetic alterations in *TP53* across various cancers. This graph illustrates the prevalence of several different types of *TP53* genetic alterations, including deep deletions, structural variants, and multiple alterations, across a range of cancer types. Data were retrieved from cBioPortal for Cancer Genomics [92]. While *TP53* is mutated at high frequency across many cancers, there are also cancers such as cervical cancer, leukemia, renal clear cell carcinoma, and thyroid cancer which display low rates of *TP53* alterations.

**Figure 2 cancers-17-00626-f002:**
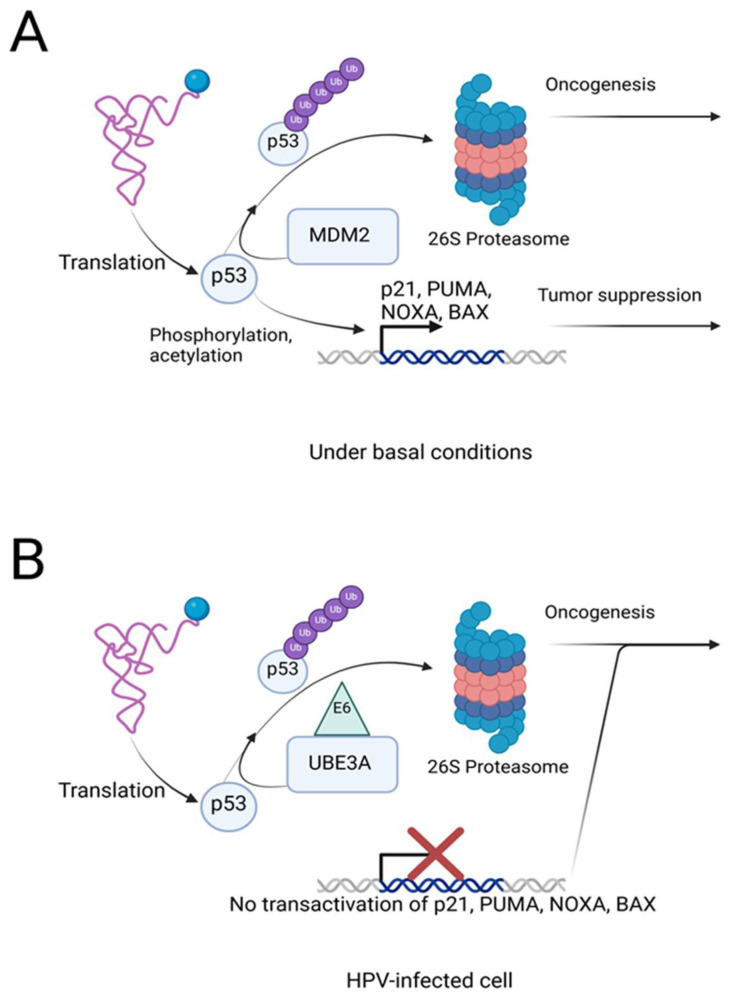
MDM2 controls the physiological turnover of the p53 protein, whereas UBE3A mediates the constitutive degradation of p53 when in complex with the HPV E6 oncoprotein. (**A**) Under basal conditions, p53 is translated and undergoes post-translational modifications such as phosphorylation and acetylation, which enhance the stability and activity of this tumor suppressor. Activated p53 mediates transcription of downstream tumor-suppressive genes including p21, PUMA, NOXA, and BAX. MDM2 is the primary regulator of the basal turnover of p53 by ubiquitination and subsequent proteasomal degradation, thereby maintaining homeostasis. Not shown here is the autoregulatory feedback between p53 and MDM2—see Section 2.1. (**B**) In HPV-infected epithelial cells, the viral oncoprotein E6 complexes with the E3 ligase UBE3A (E6AP) to target p53 for ubiquitination and degradation, independently of the MDM2-p53 axis. This results in the loss of p53-dependent transcription of tumor-suppressive genes, leading to oncogenesis.

**Figure 3 cancers-17-00626-f003:**
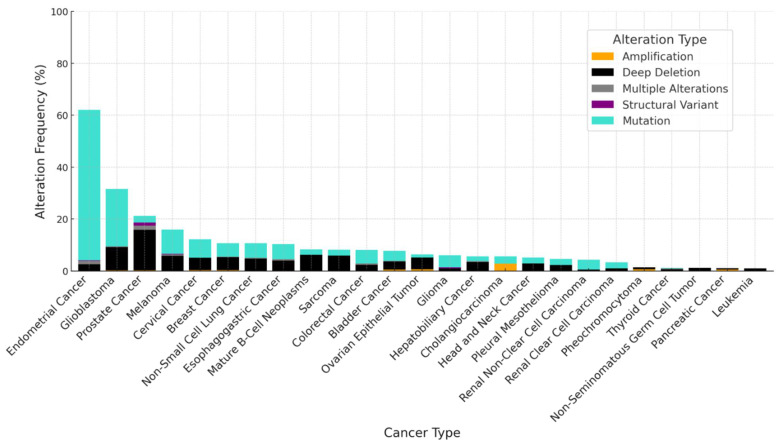
Frequency of genetic alterations in *PTEN* across various cancers. This graph highlights the various genetic alterations (mutations, deep deletions, structural variants, multiple alterations) present in *PTEN* over many types of cancer. Compared with the high frequency of *TP53* alterations (Figure 1), *PTEN* alterations are generally less frequent in many cancer types. However, the patterns of alteration between *TP53* and *PTEN* vary significantly; notably, certain cancers exhibit a higher frequency of *PTEN* deletions compared with *TP53* mutations (i.e., prostate cancer, cervical cancer, mature B-cell neoplasms). It should be noted that homozygous *PTEN* mutations are uncommon, providing support for wild-type PTEN stabilizing therapies. Data were retrieved from cBioPortal for Cancer Genomics [92].

**Table 1 cancers-17-00626-t001:** Oncogenic E3 ubiquitin ligases upstream of representative tumor suppressors. E3s were initially screened using the Ubibrowser database for E3–substrate interactions, followed by literature review for their oncogenic actions. Representative oncogenic E3s that are known to regulate TS proteins are listed.

TS Gene Name (Protein Name)	E3 (Protein)	References
*TP53* (p53)	MDM2, UBE3A/E6AP, HUWE1, COP1, PIRH2	[31,32,33,34,35,36]
*PTEN* (PTEN)	NEDD4/NEDD4-1, WWP1, WWP2, XIAP, FBXO22, RNF126, SMURF1, SPOP, STUB1/CHIP, TRIM25, TRIM27, TRIM59	[37,38,39,40,41,42,43,44,45,46,47,48]
*CDKN1B* (p27^Kip1^)	SKP2, KPC1, RNF6, COP1, PIRH2, RFWD2, TRIM21, UBE3A/E6AP, WWP1, SAG, SIAH2	[49,50,51,52,53,54,55,56,57,58,59]
*FBXW7* (FBWX7)	FBXO45	[60]
*PML* (PML)	KLHL20, UHRF1, UBE3A/E6AP, RNF111/Arkadia, R4F4, RNF8, SIAH1, SIAH2, WDR4	[61,62,63,64,65,66,67,68,69]
*BRCA1* (BRCA1)	FBXO44, HERC2, HUWE1	[70,71,72]
*RB1* (Rb)	MDM2, SKP2, UBE3A/E6AP	[73,74,75]
*TSC2* (Tuberin)	FBXW5, HERC1, PAM, UBE3A/E6AP	[76,77,78,79]

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
