# Peer review of "A Perspective on Therapeutic Targeting Against Ubiquitin Ligases to Stabilize Tumor Suppressor Proteins"

_cancers, 2025, doi:10.3390/cancers17040626_

Round 1
Reviewer 1 Report
Comments and Suggestions for Authors
This review provides a comprehensive overview of the role of E3 ubiquitin ligases in the regulation of tumor suppressor (TS) proteins and their potential as therapeutic targets in cancer treatment. The authors effectively highlight the critical distinction between the therapeutic strategies that focus on oncogene inhibition versus those aimed at restoring TS function, emphasizing the challenges associated with the latter. The discussion on the advantages of E3 ligase inhibitors over traditional proteasomal inhibitors is particularly insightful, as it underscores the potential for greater selectivity and efficacy in enhancing TS protein expression. The inclusion of recent advancements in E3 inhibitor development, such as MDM2 inhibitors, adds significant value to the narrative, illustrating the evolving landscape of targeted cancer therapies. However, I have several following concerns:
-
Clarification of Mechanisms: While the review discusses the general role of E3 ligases in TS protein degradation, a more detailed exploration of the specific mechanisms by which these ligases regulate individual TS proteins would enhance the reader's understanding of the nuances involved.
-
Clinical Implications: It would be beneficial to include a section on the current clinical trials or studies involving E3 ligase inhibitors and their outcomes. This would provide a clearer picture of the translational potential of these inhibitors in precision cancer therapies.
-
Future Directions: A discussion on the future directions for research in E3 ligase inhibition, including potential challenges and strategies to overcome them, would be valuable for guiding subsequent studies in this area.
- Abbreviations should be defined when they first appear in the text. Such as "MDM2" in Line 20, "FDA" in Line 47, ...please double check all the text and revsie them.
- The nucleic acid sequences (including gene names, regulatory sequences, and primer names) should be in italics.
- Please use the standard three-line form for the tables in the manuscript.
- If Fig.2 is from other literature or software production, please provide copyright permission.
- Please unify the format of references in the article, including the author's name, the case of words in the title of the article, the writing of the name of the journal, and the page number.
- Moderate editing of English language required.
Overall, this review is a timely and important contribution to the field, offering a promising perspective on the therapeutic potential of E3 ligase inhibitors in cancer treatment. With major revisions, it could serve as a pivotal resource for researchers and clinicians alike.
Comments on the Quality of English Language
The English could be improved to more clearly express the research.
Author Response
Reviewer 1:
Comments 1. Clarification of Mechanisms: While the review discusses the general role of E3 ligases in TS protein degradation, a more detailed exploration of the specific mechanisms by which these ligases regulate individual TS proteins would enhance the reader's understanding of the nuances involved.
Response 1.
While this manuscript is focused primarily on the biological and therapeutic outcomes of inactivating the E3s functioning upstream of TS proteins, we agree with this reviewer that a bit more mechanistic insight into the physiological E3-TS axes would be helpful for readers. Therefore, we have added descriptions of regulatory mechanisms with which the representative E3s control the TS proteins, especially the involvement of E3 or TS phosphorylation in the ubiquitination processes.
Regarding MDM2-mediatd control of p53, we have added the following sentence to the Section 2.1:
“While p53 phosphorylation is not prerequisite for MDM2-mediated ubiquitination, phosphorylation of MDM2 itself by the checkpoint kinase ATM inhibits MDM2 activity, promoting p53 stabilization in response to DNA damage [99].”
Regarding the UBE3A-p53 axis, we have added the following sentence to the Section 2.5.1:
“The E6-UBE3A-p53 ternary complex formation is sufficient for p53 ubiquitination, without involving phosphorylation of the E3 or the substrate.”
About the SKP2-p27 axis, we have added the following sentence and citation to the Section 4.:
“SKP2 plays a dominant role in temporal regulation of p27 in the cell cycle, mediating p27 ubiquitination upon CDK2-dependent phosphorylation of this CKI at the threonine-187 residue [208].”
Regarding the FBXO45-FBXW7 axis, our original manuscript discussed the PLK-2-dependent control of FBXW7 phosphorylation and ubiquitination in the Section 5, as follows
“Interestingly, FBXO45 can indirectly upregulate Polo-like kinase 2 (PLK-2), which in turn phosphorylates FBXW7 to facilitate its ubiquitination and degradation [225]. Thus, inhibiting FBXO45 would disrupt PLK-2 driven autoubiquitination of FBXW7, further stabilize FBXW7 and enhance mitotic cell death induced by the spindle toxins.”
Finally, about the KLHL20-PML axis, we have added the following sentence and citation to the Section 6:
“KLHL20-mediated PML ubiquitination depends on phosphorylation of PML by CDK1 or CDK2 and subsequent prolylisomerization by PIN1 [69].”
Comments 2. Clinical Implications: It would be beneficial to include a section on the current clinical trials or studies involving E3 ligase inhibitors and their outcomes. This would provide a clearer picture of the translational potential of these inhibitors in precision cancer therapies.
Response 2.
MDM2 inhibitors are the only category of E3 inhibitor drugs that have so far reached the stage of clinical trials. There is an excellent review published in 2024 [reference 85], which provides comprehensive information about the status and outcomes of clinical trials with MDM2 inhibitors. Therefore, we added a paragraph in Section 2.3 to clearly state the status of clinical trials with E3 inhibitors, as follows:
“MDM2 inhibitors are the only category of E3 inhibitor drugs that have hitherto reached the stage of clinical trials. An excellent review published in 2024 by Wang et al. comprehensively summarizes the status of clinical trials with MDM2 inhibitors across various stages, as well as the outcomes of closed clinical trials [85]. Here we will briefly discuss the inhibitors farthest along in clinical trials.”
Comments 3. Future Directions: A discussion on the future directions for research in E3 ligase inhibition, including potential challenges and strategies to overcome them, would be valuable for guiding subsequent studies in this area.
Response 3. According to this helpful comment, we have extensively revised the final Section 7 with a new subtitle, “Conclusion and future directions beyond the technical difficulties in targeting E3 ligases. There we have discussed numerous concerns raised from previous failures in the development of E3 inhibitors and technical challenges in targeting E3s as an emerging therapeutic strategy to reactivate TS proteins in cancer (also refer to the Response 1 for Reviewer 2). We have also highlighted technological innovations that facilitate the development of better E3-targeting therapies and clinical/molecular conditions for successful applications of such treatments.
Comments 4. Abbreviations should be defined when they first appear in the text. Such as "MDM2" in Line 20, "FDA" in Line 47, ...please double check all the text and revsie them.
Response 4. We have edited the document accordingly.
Comments 5. The nucleic acid sequences (including gene names, regulatory sequences, and primer names) should be in italics.
Response 5. We have edited the document accordingly.
Comments 6. Please use the standard three-line form for the tables in the manuscript.
Response 6. The Table 1 has been modified accordingly.
Comments 7. If Fig.2 is from other literature or software production, please provide copyright permission.
Response 7. Fig. 2 is not from other literature. We generated it for this manuscript.
Comments 8. Please unify the format of references in the article, including the author's name, the case of words in the title of the article, the writing of the name of the journal, and the page number.
Response 8. We have used the EndNote style file provided by the MDPI editorial office and the bibliography now looks consistent with the format.
Comments 9. Moderate editing of English language required.
Response 9. Someone with a graduate degree in education and communications has helped us revise the manuscript.
Reviewer 2 Report
Comments and Suggestions for Authors
This is a comprehensive review of the use of E3 ubiquitin ligases as a therapeutic target for various cancer types. Overall, it is a fantastic summary of the available literature.
There is one major overarching point. The authors should perhaps enhance their discussion of the difficulty of targeting tumor suppressors and while there is promise, there is also been a number of failures. The article is very positive about the potential of this therapeutic target and while they do cover some of the limitations, they might want to cover more of the issues of why things are and may fail.
The first sentence of the Simple Summary needs some work. As written it is difficult to follow. The rest of the review is pretty well written so this seems to be an oversight but it is a tough way to start the work.
Author Response
Reviewer 2:
Comments 1. There is one major overarching point. The authors should perhaps enhance their discussion of the difficulty of targeting tumor suppressors and while there is promise, there is also been a number of failures. The article is very positive about the potential of this therapeutic target and while they do cover some of the limitations, they might want to cover more of the issues of why things are and may fail.
Response 1. We appreciate this constructive and helpful advice from the reviewer and have extensively revised the Section 7 as “Conclusion and future directions beyond the technical difficulties in targeting E3 ligases”. The revised section addresses numerous previous failures in E3 targeting and major concerns in the field with a couple of citations, as follows:
“Since the early 2000s, targeting E3 ubiquitin ligases that control the stability of their substrate TS proteins has been evaluated as an emerging strategy to treat a variety of cancers [233]. In 2005, Garber wrote an article describing the “stall” in E3 inhibitor development [234]; nearly two decades later, we have yet to see the first bona fide FDA-approved E3 inhibitor. A major difficulty in developing therapies to target the E3-TS axes is due to the lack of well-defined catalytic domains in most E3s. For instance, RING E3s which form a vast majority of ubiquitin ligases in the proteome do not have catalytic cysteine residues, unlike HECT E3s, and mediate the ubiquitination of their substrates primarily through its PPIs with substrates and E2-ubiquitin complexes [235]. Thus, the development of PPI inhibitors is feasible only when a druggable E3-TS binding interface is present and well defined structurally. Moreover, two other major issues that are delaying the introduction of E3 inhibitors to the clinic are low efficacy and dose-limiting toxicities. Indeed, these are the primary challenges next-generation MDM2 inhibitors have faced in clinical trials. Nonetheless, technological advances such as high-resolution crystal structure studies, novel in-silico molecular modeling techniques and methodologies uncovering E3-substrate profiles have reinvigorated the development of E3-targeting therapeutics.”
Comments 2. The first sentence of the Simple Summary needs some work. As written it is difficult to follow. The rest of the review is pretty well written so this seems to be an oversight but it is a tough way to start the work.
Response 2. According to the comment, we have revised the Simple Summary, as shown below with the first two sentences:
“During cancer development, tumor suppressor proteins undergo inactivation due to genetic, epigenetic, or post-translational alterations. While the concept of reactivating tumor suppressor proteins has been appreciated as a potential anti-cancer therapeutic strategy for decades, progress in developing therapies that target tumor suppressor proteins has lagged behind the successful development of therapies targeting oncoproteins such as protein kinases.”